# Drug Repurposing Using Gene Co-Expression and Module Preservation Analysis in Acute Respiratory Distress Syndrome (ARDS), Systemic Inflammatory Response Syndrome (SIRS), Sepsis, and COVID-19

**DOI:** 10.3390/biology11121827

**Published:** 2022-12-15

**Authors:** Ryan Christian Mailem, Lemmuel L. Tayo

**Affiliations:** 1School of Chemical, Biological, and Materials Engineering and Sciences and School of Graduate Studies, Mapúa University, Manila City 1002, Philippines; 2School of Health Sciences, Mapúa University, Manila City 1002, Philippines

**Keywords:** bioinformatics, cytokine storm, drug repurposing, gene expression, genetics, systems biology, computational biology

## Abstract

**Simple Summary:**

The COVID-19 pandemic has caused a global standstill. The advent of vaccines has significantly hindered viral transmission, paving the way back to normalcy. However, post-COVID-19 treatment is still of utmost concern as people continue to suffer from “cytokine storms” or excessive inflammation, leading to death. In this paper, we utilize computational biology or bioinformatics techniques to identify possible drugs that may be repurposed to treat COVID-19 symptoms. We do so by identifying the genetic similarities and differences in three symptoms strongly associated with COVID-19: acute respiratory distress syndrome (ARDS), systemic inflammatory response syndrome (SIRS), and sepsis. As these diseases are related by the fact that they affect the immune system, similarities in their mechanisms may be exploited to find potential therapeutics. Using such an approach, we identify seven possible repurposable drugs that have been previously shown to treat immune-system related ailments. Our findings may potentially aid in driving drug design and discovery towards more effective post-COVID-19 therapeutics, and show how computational techniques can hasten the otherwise lengthy process of bringing drugs into clinical use.

**Abstract:**

SARS-CoV-2 infections are highly correlated with the overexpression of pro-inflammatory cytokines in what is known as a cytokine storm, leading to high fatality rates. Such infections are accompanied by SIRS, ARDS, and sepsis, suggesting a potential link between the three phenotypes. Currently, little is known about the transcriptional similarity between these conditions. Herein, weighted gene co-expression network analysis (WGCNA) clustering was applied to RNA-seq datasets (GSE147902, GSE66890, GSE74224, GSE177477) to identify modules of highly co-expressed and correlated genes, cross referenced with dataset GSE160163, across the samples. To assess the transcriptome similarities between the conditions, module preservation analysis was performed and functional enrichment was analyzed in DAVID webserver. The hub genes of significantly preserved modules were identified, classified into upregulated or downregulated, and used to screen candidate drugs using Connectivity Map (CMap) to identify repurposed drugs. Results show that several immune pathways (chemokine signaling, NOD-like signaling, and Th1 and Th2 cell differentiation) are conserved across the four diseases. Hub genes screened using intramodular connectivity show significant relevance with the pathogenesis of cytokine storms. Transcriptomic-driven drug repurposing identified seven candidate drugs (SB-202190, eicosatetraenoic-acid, loratadine, TPCA-1, pinocembrin, mepacrine, and CAY-10470) that targeted several immune-related processes. These identified drugs warrant further study into their efficacy for treating cytokine storms, and in vitro and in vivo experiments are recommended to confirm the findings of this study.

## 1. Introduction

The 2019 novel coronavirus, now dubbed SARS-CoV-2, has led to a global pandemic as declared by the World Health Organization (WHO) on 11 March 2020. Many studies have described the complex immune response associated with viral infection, leading to the identification of several clinical and immunological features [1,2]. It shares its mechanism of viral entry with other viruses of the *Coronaviridae* family as it is mediated by the spike (S) glycoprotein which binds to angiotensin converting enzyme 2 (ACE2) receptors that are localized in a variety of cell types, such as in the heart, liver, kidney, but most abundantly in the lungs and respiratory system (i.e., alveolar epithelial cells and capillary endothelial cells), leading to a wide range of various symptoms experienced by COVID-19 patients [3]. Despite this variance, it is of note that SARS-CoV-2 infections lead to systemic inflammatory response syndrome (SIRS) and acute respiratory distress syndrome (ARDS) [4], two relevant conditions highly correlated with COVID-19 infections [5]. Specifically, COVID-19 infections are accompanied by an overaggressive inflammatory response corresponding to a significant release of pro-inflammatory cytokines known as a cytokine storm. Though the biological relevance in relation to COVID-19 of cytokine storms has been questioned [6], studies have shown significant elevations of pro-inflammatory cytokines in COVID-19 infections as compared to other viral infections [1,2], highlighting that virulence may be more immune response-mediated rather than solely due to the viral titer. As the mechanisms of the cytokine storm have yet to be fully elucidated, there is a need to understand its genomic origin by comparing it known conditions that occur during COVID-19, such as SARS and ARDS.

With the advancement of high-throughput techniques, transcriptional profiling has proven to be an effective approach to elucidating biomarkers and drug targets, as well as identifying underlying processes and functions specific to certain diseases. In the case of COVID-19, studies have most often utilized differential gene analysis and limited research is available on clusters of highly correlated genes. Assessing the differences in gene expression between experimental and control conditions often assumes that expression levels change independently of each other, which is not the case in many biological systems in which complexity arises from gene-protein interactions as seen from [7]. Thus, a complementary approach is to study gene-gene relationships. Weighted gene co-expression network analysis (WGCNA) is a bioinformatics method for exploring the relationships between different gene sets (i.e., modules) by describing the correlation patterns between genes across multiple microarray datasets [8]. This method makes use of clustering techniques and computational analysis of module eigengenes in order to relate modules to one another. Literature shows that it has successfully been applied to construct gene co-expression networks in various diseases, most notably in cancer, to identify novel biomarkers and therapeutic targets [9]. In the case of COVID-19, where the variance in symptoms and the unclear mechanisms of the cytokine storm led to challenges in developing therapeutics to treat affected patients, the need arises to solve these challenges through relation of the COVID-19 induced cytokine storm to similar ailment (SIRS, ARDS, and sepsis), to connect their mechanisms and therefore elucidate potential novel drugs.

Adding on to this, new drug discovery is faced with the limitation of substantial costs, changing regulatory requirements, and above all, length of development and clinical testing, prior to getting government approval, making it a less desirable choice in the pharmaceutical industry. Drug repurposing (i.e., drug repositioning) is one good avenue where existing clinical drugs are applied to other diseases through relating their effects on gene expression by using bioinformatic prediction and machine learning. Compared to the development of novel drugs, drug repurposing could accelerate the translation of information regarding human diseases into therapeutic advances, all the while bypassing the lengthy and costly drug development process. Omics data, in particular, provides a useful way of relating genetics to pharmacology. The Connectivity Map (CMap) was developed as a database of over 1,000,000 expression profiles, genes, drugs, and disease states allowing connection of these variables by common gene expression signatures [10]. It has allowed for the repurposing of drugs based on gene expression values in neurogenerative diseases, cancer, and COVID-19 [11]. An integration of drug repurposing with systems biology is through module preservation analysis to identify module hub genes preserved across different diseases such as SIRS, sepsis, burns, and severe trauma [12]. These module hub genes serve as inputs into Connectivity Map analysis to identify repurposed drugs. 

Herein, a bioinformatics and systems biology approach are adopted to study the similarities and differences between the whole blood transcriptome in ARDS, SIRS, sepsis, and COVID-19, to highlight possible reusable (i.e., repurposed) drugs. Microarray datasets (GSE147902, GSE66890, GSE74224, GSE152641) from ARDS, SIRS, sepsis, and COVID-19 whole blood transcriptomes were taken from the GEO database and analyzed using WGCNA through hierarchal clustering to identify significantly co-expressed modules or clusters of co-expressed genes across the aforementioned conditions. These ailments have been widely reported to be symptoms of COVID-19 infections, and often occur in conjunction with the so-called cytokine storm [5,13]. Although the transcriptomics of COVID-19 have been widely studied [14,15,16], the specific mechanisms of cytokine storms have yet to be fully understood. The similarities between ARDS, SIRS, sepsis, and COVID-19 were analyzed at the module level using module preservation analysis and functional and pathway enrichment were used to understand their biological significance. Only the genetic similarities between the diseases were considered in this study, and temporal effects or variations were not accounted for. It is also important to note that the findings in this study are based on generated gene co-expression and PPI networks built from relating immune illnesses, and hence, offer insights into repurposed drugs to treat post-COVID-19 effects (i.e., cytokine storms). The hub genes within the highly preserved modules across the four phenotypes were determined and connectivity map analysis and database screening were used to identify possible candidate repurposed drugs.

## 2. Materials and Methods

### 2.1. Data Acquisition 

#### Gathering and Preparation of Microarray Datasets

Microarray data was sourced from the NCBI GEO [http://www.ncbi.nlm.nih.gov/geo (accessed on 24 January 2022)] database, a public repository of high throughput genomics data. A total of four datasets were used and analyzed in this study.

ARDS microarray data was taken from GSE147902 which contains the blood transcriptome of 96 patients within 24 h of ARDS onset [17], and from GSE66890, which contains the blood transcriptome of critically-ill patients with sepsis and sepsis-related ARDS [18]. SIRS and Sepsis microarray data were taken from GSE74224 which contains the blood transcriptome of 74 sepsis and 31 SIRS patients [19]. Finally, COVID-19 microarray data was taken from GSE177477 which contains the blood transcriptome of 29 COVID-19 patients and 18 healthy controls [20]. Probe identifiers of each dataset were converted to their corresponding gene symbols based on the respective probe lists of the platforms used obtained from the Affymetrix website. [http://www.affymetrix.com/ (accessed on 24 January 2022)]. Table 1 presents the summary of the characteristics of the datasets used in this study.

### 2.2. Weighted Gene Co-Expression Network Analysis (WGCNA)

#### 2.2.1. Data Input, Cleaning, and Pre-Processing

Raw data from the GEO datasets were pre-processed and normalized using the Robust Multichip Average algorithm in the “affy” R package within Bioconductor [https://www.bioconductor.org/packages/release/bioc/html/affy.html (accessed on 24 January 2022)]. Probes corresponding to controls were removed from the datasets. For the sepsis dataset (GSE66890), five of the datasets marked as excluded were removed from the analysis as per the source study [18]. The goodSampleGenes function of WGCNA was also utilized to filter out any genes and samples with too many missing values [8]. Preliminary clustering is also performed to check for any outliers. To minimize memory requirements, probe set level expression values were filtered using the coefficient of variation (CV) wherein only the highest CV probe for genes with multiple probes was retained. Additional filtering was done by only considering genes with average expression intensities of greater than 100; by removing data with lowest 20% percentile mean expression value shared by all samples; and by removing data with lowest 20% percentile variance across samples. Rows with empty gene symbols were removed, only the row with the largest mean expression value was kept when gene symbols are duplicated. Also, prior to network construction, expression values were log_2_ normalized.

#### 2.2.2. Network Construction

In module preservation analysis, one dataset serves as the reference upon which other expression data is compared. In this study, the ARDS dataset with 96 samples was used as the reference and was therefore subjected to WGCNA. Co-expression network construction and consequent module identification was performed using the WGCNA R package according to literature [8]. Briefly, pre-processed data was input into R. Soft thresholding power was determined using the pickSoftThreshold function which analyzes the network topology to find the best power from a range of 1–20 based on the “scale-free topology criterion” of which many biological systems exhibit. Network type was set to signed, minimum module size was set to 30, and medium sensitivity (deepSplit) were set to 2. This results in an adjacency matrix.

The adjacency matrix was transformed into a network of Topological Overlaps (TO) which quantifies the co-expression relationships considering all pairs of genes in relation to all other genes in the network. Hierarchical clustering was then applied using 1–TO as the distance measure, resulting in a cluster dendrogram where the branches correspond to gene co-expression modules. This is known as the Dynamic Tree Cutting algorithm as described in [21]. The resulting groups of genes (i.e., modules) are representative of genes with consistently correlated expression profiles across the sample datasets.

#### 2.2.3. Module Preservation Analysis

After construction of the weighted gene co-expression network of the ARDS dataset whose modules served as reference, the sepsis, SIRS, and COVID-19 datasets were projected onto the reference to analyze module preservation. This was also done using the WGCNA package through the modulePreservation function [8]. The parameters for module presentation were set as follows: the number of permutations (nPermutations) was set to 100 and the network type (networkType) was set to signed. The module qualities were also assessed and plotted in R.

### 2.3. GO Enrichment and KEGG Pathway Analysis of Module Genes

Genes within the identified modules of interest were extracted from the network and functional enrichment was performed using the Database for Annotation, Visualization, and Integrated Discovery (DAVID) webserver (accessed on 26 July 2022), which interprets and correlates the function of various genes based on previous literature [22]. Specifically, functional annotation was performed using the Gene Ontology (GO) database, a repository of the basic functions and characteristics of genes and their protein products, while the Kyoto Encyclopedia of Genes and Genomes (KEGG) database was used to infer information regarding the biological, genomic, chemical, and systemic functions of various genes [23,24]. Only those genes present with functional significance and those present within the databases were included for the analysis. Results with *p* < 0.05 were considered to be statistically significant when screening for GO terms and KEGG pathways.

#### Identification of Hub Genes within Modules

The constructed gene co-expression network was visualized using Cytoscape [25]. Hub genes within the identified modules were identified through the inter-connectivity value. Briefly, for each identified module, the module eigengene based connectivity (kME) was calculated using the “intramodular Connectivity” function from the WGCNA package [8]. Genes with the highest intra-modular connectivity values were considered to the hub genes. 

Additionally, to account for non-protein coding genes, the genes of each preserved module were passed through the STRING database to construct a protein-protein interaction (PPI) network. The resulting PPI network was imported into Cytoscape and the hub genes were selected using cytoHubba. Degree, Maximum Neighborhood Component (MNC), Radiality Centrality, Stress Centrality, and Closeness Centrality were the algorithms used to evaluate the identified hub genes. Consistent genes were taken to be hub genes. The top three hub genes were taken for further downstream analysis. 

### 2.4. Screening of Possible Repurposed Drug Candidates

#### Connectivity Map Analysis

To screen possible repurposed drug candidates that are associated with the preserved modules across ARDS, SIRS, sepsis, and COVID-19, connectivity map analysis was performed using the Connectivity Map online webserver [10]. Specifically, the top three genes were taken from each of the identified modules preserved in COVID-19. Each of the extracted genes were grouped into either up-regulated or down-regulated. The two groups of genes were submitted to Connectivity Map through the CLUE.io webserver. The significance threshold value was set to 0.01 and only the candidates with negative enrichment scores were considered. Connectivity scores range from +100 to −100 (from completely positive correlation to completely negative). The more negative the correlation, the greater the drug reverses the inputted upregulated genes and as such, means that the drug is able to lessen the expression of the hub genes used in this study. For simplicity, the top five drugs identified were recorded.

## 3. Results

### 3.1. WGCNA Network Construction

Prior to network construction, the datasets were normalized and filtered, where samples with excessive missing values were removed. Clustering was then performed and the results are shown in Figure 1. No significant outliers were identified based on the generated dendrogram and as such the data was deemed fit for network construction.

The ARDS dataset was used for the construction of the reference gene co-expression network. The dataset had a total of 96 samples which were considered for construction. All datasets were gene expression profiles obtained from the whole blood samples of individuals within 24 h of ARDS onset. After filtering according to the criteria discussed in the methodology, the log10 normalized expression values of 19,253 genes were used to construct the network.

Gene co-expression modules were identified by WGCNA. Through checking the independence of the network on the scale, the softPower threshold of 18 was determined to be the most appropriate (Figure 2, left), as also reflected by the maximum mean connectivity (Figure 2, right). Setting the softPower threshold to this value led to an approximately scale-free network topology which is a common characteristic for biological networks (Figure 3, right). The values used to compute for the linearity are presented as a histogram (Figure 3, left).

Using the calculated softPower thresholds, a scale-free topology is obtained (see Figure 3, right). This suggests that the genes under consideration in this study, at least statistically, are biologically relevant and are suitable for co-expression analysis. WGCNA was able to identify a total of 46 gene co-expression modules. These modules may have identified genes with significantly similar with significantly similar expression profiles. Co-expression similarities of the entire modules was quantified through calculation of their eigengenes and clustered based on their correlation, resulting in 13 merged gene co-expression modules from the initial 46. The identified gene modules before and after clustering and merging are shown in the cluster dendrogram in Figure 4. The grey module represents unassigned genes which were not considered in subsequent downstream analysis.

Each identified module was assigned a color. The turquoise module had a total of 2803 genes; the magenta module had 410 genes; the light cyan module had 915 genes; the light green module has 265 genes; the light yellow module has 455 genes; the orange module has 123 genes; the sky blue module had 100 genes; the violet module had 94 genes; the dark olive green module had 91 genes; the yellow green module had 87 genes; the floral white module, which was the largest, had 6699 genes; the dark orange module had 120 genes; and finally, the brown module had 39 genes.

### 3.2. Gene Ontology (GO) Enrichment Results

To understand the biological relevance of the identified modules, the genes were submitted to the DAVID and gProfiler webservers for functional GO enrichment. The RefSeq and Genbank IDs for each gene were submitted to gProfiler and converted to their Entrez IDs. Genes without a specific entrez ID were labeled with NONE and not considered for functional enrichment as they would not be found in any available database. For simplicity, only the top enriched GO terms were recorded for each module and presented in Figure 5A–C, for biological process (BP), molecular function (MF), and cellular component (CC), respectively. More detailed information can be found in Appendix A. The hub genes shown are based on the genes with the highest intra-connectivity score as computed by WGCNA.

All modules were significantly enriched with a specific BP, MF, and CC. Of relevance to the topic of this paper are the upregulated M9, M18, M28, M32, and M36, all of which were observed to be enriched in biological processes directly linked to the immune response (Figure 5A–C). They were also observed to be enriched in cellular components related to immune cells, such as M9 being enriched in the T-cell receptor complex. Interestingly, the two largest modules, M44 and M1, were enriched in large-scale macromolecule metabolic processes (2295/5606) and chemical stimulus detection (399/1583) (Figure 5A), cellular processes which have been observed to be affected by COVID-19. In terms of molecular function, M44 had 4893 genes involved in protein binding and binding. Small modules (those with sizes < 50) were also found to be significantly enriched in a specific BP, MF, and CC. For example, the M46 was found to be enriched in genes involved in spermatogenesis (GO:0007283). However, only a small percentage of the genes in the M46 were found to be involved in the process. This is due to the small size of the original query. Further, modules were also observed to be involved in multiple immune-related molecular functions. M9 genes were observed to be enriched in peptide antigen binding; M28 genes were enriched in antigen binding; and M32 genes were involved in MHC Class II receptor activity. Collectively, the identified modules were significantly enriched in various BPs, MFs, and CCs.

### 3.3. KEGG Pathway Enrichment

To further assess the biological relevance of the identified modules, the modules were subjected to KEGG Pathway enrichment. The results are visualized in Figure 6 by taking the most significant KEGG pathway enriched for each module. Detailed results are presented in Appendix A.

Results show that not all modules were enriched in a specific KEGG pathway. Specifically, M28 and M46 were not significantly enriched in any KEGG pathway, potentially due to the low size of the modules. Other modules, however, were enriched in varying pathways. For example, M1 was enriched in olfactory transduction, similar to its observed enriched GO term; M9 genes played a role in Th1 and Th2 cell differentiation which is linked to its role in the adaptive immune response and the T-cell receptor complex; and M16 genes were involved in the chemokine signaling pathway; pathways linked to immune processes which suggest the genes’ involvement in ARDS, an inflammatory complication.

Other modules were also observed to be involved in KEGG pathways linked to the immune response. For example, M18, M33, and M36 were enriched in the NOD-like receptor signaling pathway (KEGG:04621), B-cell receptor signaling pathway (KEGG:hsa04662), and platelet activation (KEGG:04611), respectively. Further, some modules were involved in cell transport and differentiation, such as M44 and M332, which were involved in nucleocytoplasmic transport (KEGG:03013) and hematopoietic cell lineage (KEGG:04640), respectively.

### 3.4. Module Preservation Analysis

To compare the similarity of gene expression at the module level between the four diseases included in this study (ARDS, SIRS, Sepsis, and COVID-19), the network constructed based on the ARDS dataset was used as reference for module preservation analysis where other datasets describing SIRS, sepsis, and COVID-19 whole blood transcriptome expressions were projected onto the ARDS dataset. The results are summarized in the graph, shown in Figure 7, with each module labelled according to their assigned color. Additional statistical analysis revealed that preserved modules were all of high quality based on the summary quality statistic (Appendix A) suggesting that all the significantly preserved modules were of statistical significance.

Module preservation analysis revealed that the ARDS modules were most significantly preserved in the sepsis dataset (Figure 7B), while only a few modules were strongly preserved in the SIRS and COVID-19 datasets. In SIRS, sky blue (M28) and dark orange (M45) showed no preservation, while orange (M25), violet (M32), dark olive green (M33), and yellow green (M36) showed moderate preservation, and turquoise (M1), magenta (M9), light cyan (M16), light green (M18), light yellow (M19), and floral white (M44) showed strong preservation (Figure 7A). In sepsis, no modules showed no preservation. Turquoise (M1), sky blue (M28), violet (M32), and dark orange (M45) showed moderate preservation while magenta (M9), light cyan (M16), light green (M18), light yellow (M19), orange (M25), dark olive green (M33), yellow green (M36), and floral white (M44) showed strong preservation (Figure 5B). Finally, in COVID-19, sky blue (M28) showed no preservation, orange (M25), dark orange (M45), violet (M32), and dark olive green (M33) showed moderate preservation, and turquoise (M1), magenta (M9), light cyan (M16), light green (M18), light yellow (M19), yellow green (M36), and floral white (M44) showed strong preservation (Figure 7C). Collectively, the modules strongly preserved in COVID-19 were also strongly preserved in other modules and were therefore selected for further downstream analysis.

### 3.5. Hub Gene Identification

One of the limitations of WGCNA is that the network generated is undirected, that is, the network established only provides information on gene correlation and not on gene or protein interaction. Considering that most drugs have mechanism of actions that target proteins, the hub genes were selected by submitting the genes in each module preserved in the COVID-19 dataset to the STRING database for PPI network construction. The networks were imported into Cytoscape and cytoHubba was utilized to determine the top three hub genes by taking the three most significant genes as identified by the five algorithms, with the results being presented in Table 2. These top three hub genes were all found to be upregulated in their respective modules. The selected genes were selected for determining potential repurposed drugs.

### 3.6. Candidate Drug Identification

To identify candidate drugs, the hub genes identified in the previous section were submitted for Connectivity Map analysis using the CLUE.io webserver and the top seven genes with the lowest connectivity score were selected as candidate drugs. The identified drugs and their respective functions are summarized in Table 3. Only those drugs with connectivity scores lower than −95.00 were statistically significant as per CLUE.io’s documentation. The negative connectivity scores of each drug suggest that the drugs, upon binding or exertion of their mechanism of action, reverse the gene expression of the query, or in other words, the perturbagens identified exhibit downregulation of the identified upregulated hub genes. Results showed that SB-201290 was the drug with the lowest connectivity score, while CAY-10470 had the highest score. It can be observed that all identified drugs served to inhibit several pathways involved in the immune response.

## 4. Discussion

Correlation networks have been growing in popularity for the bioinformatics analysis of gene regulation. Weighted gene co-expression analysis (WGCNA) has been used to apply a systems biology network approach to describe gene correlation patterns across samples. Such an approach has been used to identify candidate biomarkers and therapeutic targets in cancer, mouse and yeast genetics, and the analysis of brain imaging data [8]. Further, cytokine storms have become recent issue with the rise of COVID-19 infections. Patients suffering from severe cases have been observed to present symptoms of a cytokine storm, described as the acute influx of pro-inflammatory cytokines. Despite the knowledge that cytokine storms are often accompanied by ARDS, SIRS, and sepsis, the relationship between these diseases has yet to be elucidated. Inherently, with the ARDS and SIRS in this present study, a quantitative network-based approach was employed to statistically determine the co-expressed genes across four related diseases: ARDS, SIRS and sepsis, and symptomatic COVID-19, using NCBI GEO datasets to investigate the genetic similarities between the diseases to highlight conserved and divergent pathways in cytokine storms and identify candidate repurposed drugs based on gene co-expression data. Though the study does not consider the temporal nature of the datasets, this module-based analysis is thought to give more consistent results despite the time-series nature of sepsis microarray samples as the function of module is more stable than that of a single gene.

Table 4 presents the hub genes screened via intramodular connectivity along with their corresponding functions. Several of these genes have been linked to ARDS and inflammation and consequently, cytokine storms, as highlighted in the table. Of particular interest is TREM-1, which has been previously reported to accentuate lung injury during ARDS and has shown promise as an inflammatory biomarker and as therapeutic target for sepsis and septic shock [26,27]. It has been reported to function in regulating T-cell proliferation and activation of antigen presenting cells during the antiviral immune response by amplifying toll-like receptor (TLR)-initiated responses against microbial challenges [28]. Recently, the extracellular cold-inducible RNA-binding protein (eCIRP), which is the endogenous ligand of TREM-1, was found to induce acute lung injury during sepsis and has been shown to induce inflammation in alveolar cells through induction of TREM-1 and activation of cytokine production in macrophages [29], further highlighting the protein’s role in stimulating the immune response. Considering its extensive role in regulating the expression of cytokines and that macrophages have been previously linked to symptoms reminiscent of COVID-19, it could be another potential therapeutic target for cytokine storms. These hub genes could be interesting biomarker and therapeutic target candidates for the discussed diseases and warrant further study.

Preserved gene co-expression modules were further identified via module preservation. Interestingly, two of the largest modules (M1 and M44) were significantly preserved in the COVID-19 dataset. M1 genes were found to be significantly enriched in olfactory transduction and may contain genes affected during COVID-19 infections leading to the loss of smell and taste. M44, on the other hand, was enriched in nucleocytoplasmic transport and its hub gene has been implicated in modulating cytokine production during sepsis. Several immune-related pathways were also significantly preserved in the COVID-19 dataset alluding to potential conserved mechanisms across the four diseases. M9 was enriched in Th1 and Th2 cell differentiation. T-helper (Th) cells regulate and produce IFN-γ and TNF-α, which are macrophage activators and are responsible for cell-mediated immunity. These synergism of these two cytokines has been previously shown to be sufficient in inducing cell and tissue damage reminiscent of COVID-19 symptoms [35]. More recently, an imbalance in the expression of Th1/Th2 has been associated with disease severity and outcome. Under normal conditions Th1 cells can control the infection sufficiently and has even been linked with a favorable outcome [36]. However, in severe cases, imbalances such as in cytokine storms lead to Th2 cell priming that results in poor prognosis [37]. Thus, M9 is potentially a significant module related to COVID-19 disease severity brought about by cytokine storms.

Chemokine signaling pathways were also found to be significantly preserved in symptomatic COVID-19. This pathway is initiated by ligand-receptor binding in several chemokine receptors and is responsible for activation of several downstream signaling events such as the MAPK pathway, which has been implicated in cytokine storms [38]. Several of the chemokine receptors (e.g., CXCR3 and CXCR4) have also been shown to be upregulated during cytokine storm-like symptoms as upon binding of their natural ligands (e.g., CXCL12), inflammation is promoted and chemotaxis induces apoptosis potentially leading to significant tissue damage as seen in COVID-19 cases [39]. Considering also that cytokine storms begin with the accumulation of pro-inflammatory cells in sites of severe infection, chemokines are an important driver for inflammation as they serve as chemoattractants. Their overexpression could be a possible mechanism for observed lung tissue damage (i.e., ARDS) in COVID-19 cases. This further emphasizes the importance of chemokine signaling in the movement of immune cells during a cytokine storm and serves as additional support for chemokine receptors to be potential druggable targets. Another pathway conserved in symptomatic COVID-19 is the NOD-like receptor signaling pathway. NOD-like receptors (NLRs) are essential for detecting pathogen-specific molecules and upon sensing infections, activates signaling pathways that drive NF-κB-dependent expression of pro-inflammatory cytokines.

Moving on, one of the main limitations of this study is that WGCNA generates an undirected network–that is, it simply presents gene correlation quantitatively based on statistical analysis and does not necessarily reflect any possible gene or protein interactions between genes. Hub genes identified using intramodular connectivity may not necessarily be correlated in terms of interaction, and merely serve as a basis of comparison herein. As the main goal of this paper is to be able to identify candidate repurposed drugs, to circumvent this limitation, and considering that drugs typically act on protein targets, this study submitted the identified gene co-expression modules to STRINGdb for PPI network construction to extract such interactions and to identify hub genes based on topological connectivity. Results suggested that the identified co-expressed genes were also significantly related in terms of biochemical interactions as the PPI networks generated were of high significance (*p* < 0.05). This approach allowed identification of several relevant hub genes as shown in Table 2. Some hub genes (e.g., STAT1, STAT3) were present in both methods of selection, suggesting at least partial similarity between methods. These genes, similar to the hubs identified by screening intramodular connectivity, have been closely linked with inflammation and potentially, cytokine storms [30]. Interestingly, the STRINGdb approach was able to identify several biologically relevant proteins such as TLR4 and IRF1 which have both been linked to cytokine storms with the former being responsible for induction of cytokine production and the latter having been shown to alleviate cytokine storm-like symptoms upon its downregulation in mice models [35]. The resulting PPI networks were also shown to be biologically significant statistically and be highly connected. Several of the identified pathways in the other method also remained conserved even after database screening indicating that the primary genes enriched in specific pathways remain present in the module networks. These results suggest that utilizing PPI network construction prior to CMap analysis is a good way to narrow down and specify results without loss of significant information especially when screening for possible repurposed drugs.

Another growing avenue for treating various diseases is the repurposing of already-established drugs otherwise known as drug repositioning. This study was able to utilize preserved gene modules in COVID-19 to identify such drugs. The identified drugs (see Table 3) were all found to target immune-related pathways. The MAPK pathway has been a promising target for novel therapeutic approaches against COVID-19 as the loss of ACE2 activity brought about by viral infection leads to overexpression of p38, which leads to inflammation [38]. COX-inhibitors are common in treating inflammation and has been used for treating ARDS [40]. Antihistamines alleviate symptoms of allergies through inhibition of histamine receptors which have been theorized to induce TNF-α-stimulated accumulation of pro-inflammatory cytokines [41]. Further, a study that treated 22,560 COVID-19 patients with histamine antagonists (e.g., loratadine) resulted in attenuation of cytokine release leading to moderate to significant relief of COVID-19 symptoms, suggesting the potential of antihistamines in treating COVID-19 and cytokine storms in general [42]. Additionally, IKK*β* is a protein responsible for transducing pro-inflammatory signals leading to its anti-inflammatory properties. Recently, a study has shown that nanosomal delivery of TPCA-1 to porcine chondrocytes can prevent TNF-α stimulation thereby lowering inflammation [43]. Several of the drugs identified were also found to inhibit the NF-κB pathway which is central to cytokine storms [44,45,46]. Another paper has also shown that targeted delivery of TPCA-1 using platelet-derived extracellular vesicles is able to “calm” the cytokine storm in pneumonia mice models through inhibition of pro-inflammatory factors [47]. This suggests that TPCA-1 could be a good candidate drug for treatment of cytokine storms in COVID-19. Mepacrine is a known anti-malarial drug that has since been discovered to have inhibitory activity against NF-κB and is now used as an NSAID to inhibit cytokine production. Previous studies have shown that it is effective in alleviating airway inflammation in murine models of asthma [48]. Such inflammation is also present in COVID-19 patients and thus points to its potential as a therapeutic. Recently, antimalarials have also shown some promising results for being repurposed to tackle COVID-19 cytokine storms and the resulting neuroinflammatory symptoms [49].

The identified compounds were all found to have some significance with regards to attenuating or reducing inflammatory pathways that have been previously linked to cytokine storms [39]. Some of the identified drugs have also already been utilized to treat diseases unrelated to cytokine storms, thus indicating the drug repurposing is a promising avenue for drug discovery. Combining gene expression data and a network-based systems biology approach is a good pathway for transcriptome-driven drug discovery. The results of this study suggest the importance of considering gene expression when designing drugs or active compounds and may offer insights to the repositioning of drugs. Future studies can pursue validation of the results of this study to establish the viability of this approach as well as to identify potential effective therapeutics against cytokine storms.

## 5. Conclusions

Cytokine storms have been observed to lead to ARDS, sepsis, and SIRS, which may indicate specific shared genetic mechanisms. Herein, a network-based approach was applied to correlate the gene expression patterns between the aforementioned diseases by using WGCNA and module preservation analysis to identify co-expressed genes preserved across the four diseases. SARS-CoV-2 with the D614G mutation on the spike protein was the main strain analyzed in this study. A total of 13 stable gene co-expression modules were identified from the whole blood transcriptome of ARDS patients. Modules were more strongly preserved in the sepsis dataset, and less strongly preserved in the SIRS and COVID-19 datasets. Among these identified modules, M1, M9, M16, M18, M19, and M44 were strongly preserved in COVID-19 and across all four diseases. These modules were enriched in olfactory transduction, Th1 and Th2 cell differentiation, chemokine signaling, NOD-like receptor signaling, mitophagy-animal, and nucleocytoplasmic transport KEGG pathways. Genes within each module were shown to be significantly biologically-related based on PPI network construction. Identified hub genes and pathways align with a previous study and therefore reinforce their potential as therapeutic biomarkers and druggable targets. The seven identified candidate drugs also warrant further investigation as potential therapeutics against cytokine storms. Further experiments are recommended however to verify their effectiveness against cytokine storms.

## Figures and Tables

**Figure 1 biology-11-01827-f001:**
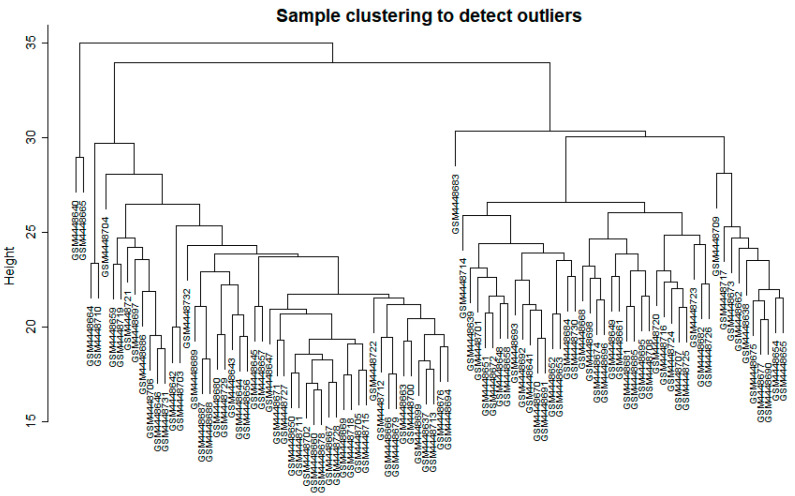
Clustering dendrogram of samples based on their Euclidean distance to detect outliers.

**Figure 2 biology-11-01827-f002:**
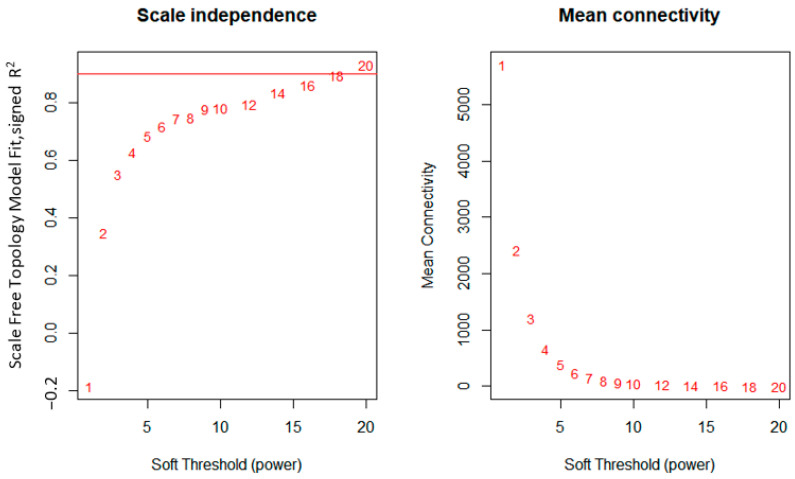
Results of determination of soft threshold power to ensure scale-free topology (**left**) and the summary of mean connectivity score at each threshold (**right**).

**Figure 3 biology-11-01827-f003:**
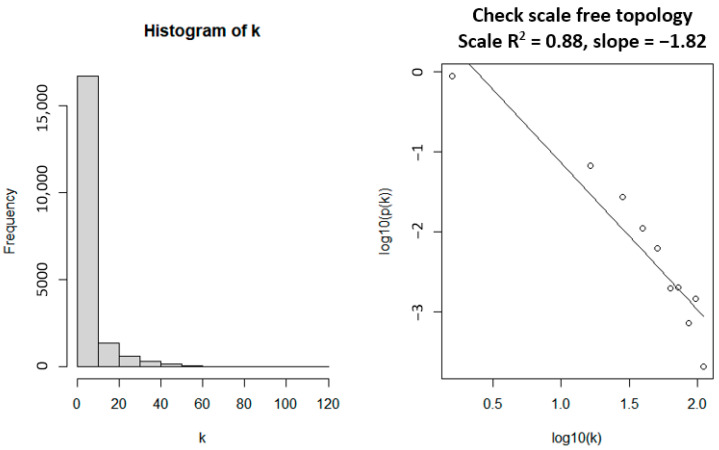
Power law distribution graph to verify scale-free topology of the network.

**Figure 4 biology-11-01827-f004:**
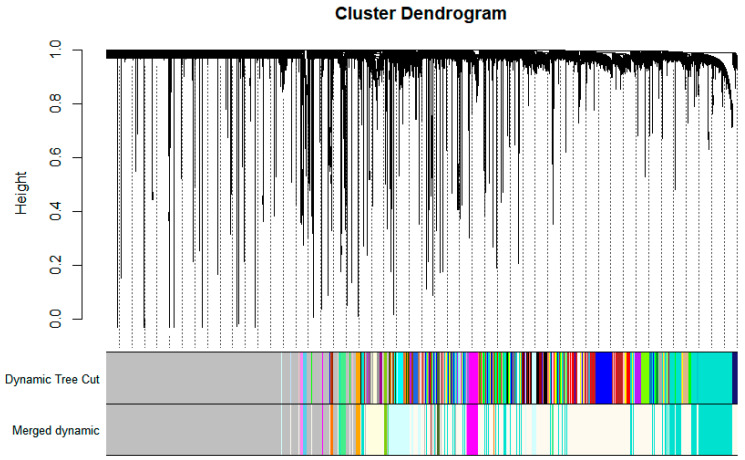
Cluster dendrogram of the fourteen identified modules from the network with each module being assigned a color. Genes in the grey module were not significantly clustered and were excluded in the analysis.

**Figure 5 biology-11-01827-f005:**
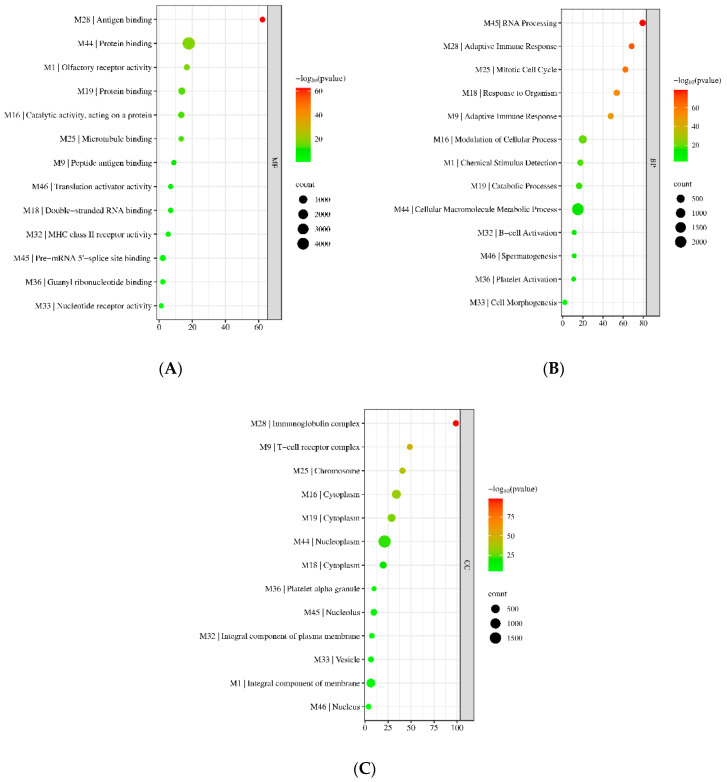
Bubble plots for the enriched (**A**) molecular functions (MF), (**B**) biological processes (BP), and (**C**) cellular components (CC) for each preserved module.

**Figure 6 biology-11-01827-f006:**
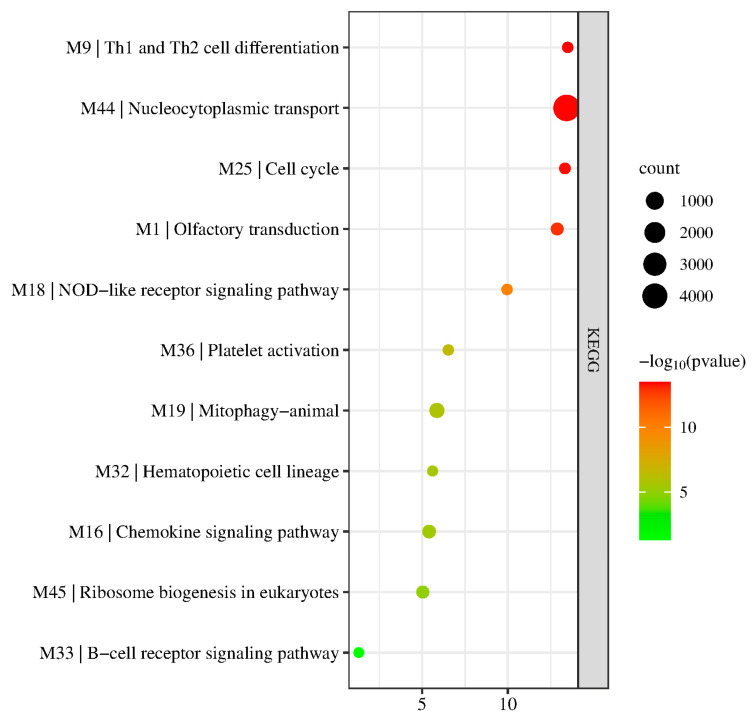
Bubble plots for the enriched KEGG pathways for each preserved module.

**Figure 7 biology-11-01827-f007:**
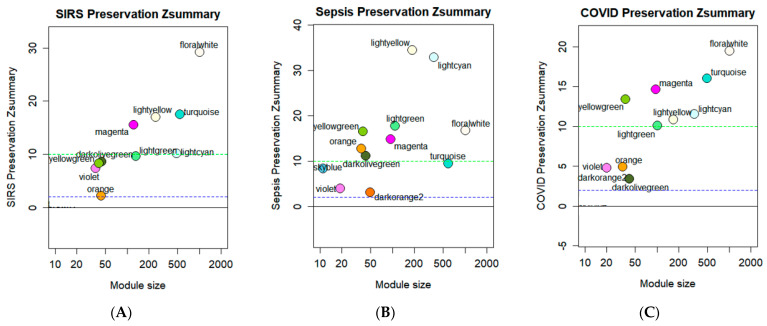
Module preservation analysis results highlighting the highly conserved co-expressed gene modules between SIRS (**A**), sepsis (**B**), and COVID-19 (**C**), using ARDS as reference. Summary statistic indicates −log10 statistical significance of each module. Green and dark blue dashed lines represent cut-off of strong preservation and moderate preservation, respectively.

**Table 1 biology-11-01827-t001:** Summary of NCBI GEO datasets utilized for WGCNA and module preservation analysis.

	2020	2015	2015	2021
	GSE147902	GSE66890	GSE74224	GSE177477
**Type**	Expression profiling by array			
**Condition**	ARDS	Sepsis	SIRS	COVID-19
**Platform**	Affymetrix Human Gene 2.1 ST Array	Affymetrix Human Gene 1.0 ST Array		Affymetrix Clariom S Assay
**Source**	Whole Blood RNA			
**No. of samples**	96	58	105	47

**Table 2 biology-11-01827-t002:** Top three hub genes of conserved modules ranked as ranked by cytoHubba after PPI network construction through stringDB.

Module	Color	Top 1	Top 2	Top3
1	Turquoise	GAPDH	AKT1	ALB
9	Magenta	CD8A	CTLA4	LCK
16	Light Cyan	CTNNB1	TLR4	STAT3
18	Light Green	STAT1	IRF7	IFIH1
19	Light Yellow	UBB	UBA52	TFRC
36	Yellow Green	GP6	PF4	ITGB3
44	Floral White	HDAC1	HSPA8	RPS3

**Table 3 biology-11-01827-t003:** Candidate drugs as identified using Connectivity Map through CLUE.io.

Disease	Rank	Name	Connectivity Score	Function
COVID-19	1	SB-202190	−99.44	P38 MAPK inhibitor
	2	Eicosatetraenoic-acid	−99.40	COX inhibitor
	3	Loratadine	−99.37	Histamine receptor antagonist
	4	TPCA-1	−99.12	IKK inhibitor
	5	Pinocembrin	−98.58	CYP1B1 inhibitor
	6	Mepacrine	−97.22	Cytokine production inhibitor
	7	CAY-10470	−96.80	NFkβ pathway inhibitor

**Table 4 biology-11-01827-t004:** Hub genes and their corresponding functions screened via intramodular connectivity.

Hub Gene	Function	Reference
TRIM49D2	Protein-coding gene predicted to play a role in the innate immune response.	
TRAJ12	Plays a role in joining the two subunits of the T-cell receptor.	
ACAP2	Enables GTPase activator activity.	
STAT2	Induces IFN immune response and has been observed to be aberrant in COVID-19 cytokine storms.	[30]
SLC1A5	Solute carrier which affects ferroptosis, a potential mechanism for tissue damage during cytokine storms.	
TOP2A	DNA topoisomerase that has been highlighted in gene expression studies of COVID-19.	[31]
IGKV1-6	Variable domain of immunoglobulin light chains that facilitates antigen recognition.	
AFF3	Transcriptional activator found primarily in lymphoid tissue.	
LRP1	Facilitates cellular movement; controls cellular cytokine signaling.	[32]
METTL23	Methyl transferase that has been reported to be upregulated in sepsis.	[33]
SNORD104	Involved in immune homeostasis.	[34]

## Data Availability

The gene microarray datasets used for the study are openly available in the NCBI Gene Expression Omnibus (GEO) database under the accession IDs GSE160163, GSE147902, GSE66890, GSE74224, and GSE177477 at https://www.ncbi.nlm.nih.gov/geo/ (accessed on 24 January 2022).

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
