# Peer review of "Drug Repurposing Using Gene Co-Expression and Module Preservation Analysis in Acute Respiratory Distress Syndrome (ARDS), Systemic Inflammatory Response Syndrome (SIRS), Sepsis, and COVID-19"

_biology, 2022, doi:10.3390/biology11121827_

Round 1

Reviewer 1 Report

In this present study, a quantitative network-based approach was employed to statistically determine the co-expressed genes across four related diseases: ARDS, SIRS and sepsis, and symptomatic COVID-19 using NCBI GEO datasets to investigate the genetic similarities between the diseases to highlight conserved and divergent pathways in cytokine storms and identify candidate repurposed drugs based on gene co-expression data. The work is important, but some points need to be clarified and also the way of presenting the data/results needs to be improved. Therefore, the article should be accepted after major revisions:

1) In figure 4 it is not clear the correlation between the number of genes and colors is. Therefore, this figure needs to be improved. Consequently, this will reflect in the colors of the modules of each table (which must be modified to facilitate the interpretation of the data - it is very confusing).

2) If the generated network is undirected, i.e., established only provides information on gene correlation and not on gene or protein interaction, how to determine each target protein of each module? This needs to be further explained in the manuscript.

3) Although drug repurposing is a promising avenue for drug discovery, the in silico search for anti-COVID drugs may favor the misuse of antiparasitic or antimicrobial drugs against a virus. The authors must make clear the limitations of this study. Furthermore, what is the direct relationship between the four diseases? Sepsis is a disease triggered by a marked systemic inflammatory response to an infection caused by bacteria. How is this disease related to COVID-19 caused by SARS-CoV-2? Authors should beware of comparisons that although they have some correlation, the origin is completely different (this needs to be discussed in the manuscript).

Author Response

Dear reviewer, thank you for your very interesting and thought-provoking inputs. We have sincerely considered your suggestions and made the appropriate changes to the manuscript. Please see the attachment for our response. 

Reviewer 2 Report

The article 'Drug Repurposing using Gene Co-Expression and Module 2 Preservation Analysis in Acute Respiratory Distress Syndrome 3 (ARDS), Systemic Inflammatory Response Syndrome (SIRS), 4 Sepsis, And COVID-19', by Mailem and Tayo, merits to be published in Biology after minor revision.

I would like to ask the authors why they did not use consensus modue analysis in the present work.

Some minor details:

- Which is the affiliation of the authors? These fields are empty.

- Is this e-mail address correct:  rcmailem@mymail.mapua.edu.ph? Maybe it should be  rcmailem@mapua.edu.ph.

- In the title, 'And COVID-19' should be 'and COVID-19'

- In line 150, the link http://www.affymetrix.com/ is broken

- In line 213, please correct 'intramodularConnectivity'

- On page 6, Figure 2, left and Figure 3, right are mentioned in the text. The authors should also comment Figure 3, left and Figure 2, right

- Figure 5 identifies the modules by color, and the corresponding text uses module numbers. This is a bit confusing, and the authors should give any solution to make life easier for readers. Maybe using colored text?

Author Response

Dear reviewer, thank you for your honest comments. Please see the attachment for our response.

Reviewer 3 Report

Research article on WGCNA approach is fine piece of work.

Few additions can be done to the article to make it easy to understand.

1.     For explaining gene ontology analysis, figures format should also be included in the figure in addition to table.

2.     Discussion in the paper is very long and is difficult to follow the functions of each gene identified with different methods. I will suggest authors to make table with several headings to explain the discussion.

3.     Some sentences are long and repetitive in the introduction which should be corrected in the revised edition of the paper.

4.     Authors should use the short form later if they have already mentioned the long form before in the beginning of the paper, like use cMAP instead of Connectivity Map.

Author Response

Dear reviewer, please see the attachment for our response.

Round 2

Reviewer 1 Report

The authors answered all questions and made the necessary changes to the text, so the manuscript should be accepted for publication in Viruses.